# Wall Teichoic Acids Facilitate the Release of Toxins from the Surface of *Staphylococcus aureus*

Tarcisio Brignoli,[a] Edward Douglas,[a,b] Seána Duggan,[a,c] Olayemi Grace Fagunloye,[d] Rajan Adhikari,[e] M. Javad Aman,[e] Ruth C. Massey[a,f,g,h]

[a]School of Cellular and Molecular Medicine, University of Bristol, Bristol, United Kingdom
[b]Biology and Biochemistry Department, University of Bath, Bath, United Kingdom
[c]MRC Centre for Medical Mycology, University of Exeter, Exeter, United Kingdom
[d]School of Medicine, University of Pittsburgh, Pittsburgh, Pennsylvania, USA
[e]Integrated Biotherapeutics, Inc. (IBT), Rockville, Maryland, USA
[f]School of Microbiology, University College Cork, Cork, Ireland
[g]School of Medicine, University College Cork, Cork, Ireland
[h]APC Microbiome Ireland, University College Cork, Cork, Ireland

**ABSTRACT** A major feature of the pathogenicity of *Staphylococcus aureus* is its ability to secrete cytolytic toxins. This process involves the translocation of the toxins from the cytoplasm through the bacterial membrane and the cell wall to the external environment. The process of their movement through the membrane is relatively well defined, involving both general and toxin-specific secretory systems. Movement of the toxins through the cell wall was considered to involve the passive diffusion of the proteins through the porous cell wall structures; however, recent work suggests that this is more complex, and here we demonstrate a role for the wall teichoic acids (WTA) in this process. Utilizing a genome-wide association approach, we identified a polymorphism in the locus encoding the WTA biosynthetic machinery as associated with the cytolytic activity of the bacteria. We verified this association using an isogenic mutant set and found that WTA are required for the release of several cytolytic toxins from the bacterial cells. We show that this effect is mediated by a change in the electrostatic charge across the cell envelope that results from the loss of WTA. As a major target for the development of novel therapeutics, it is important that we fully understand the entire process of cytolytic toxin production and release. These findings open up a new aspect to the process of toxin release by a major human pathogen while also demonstrating that clinical isolates can utilize WTA production to vary their cytotoxicity, thereby altering their pathogenic capabilities.

**IMPORTANCE** The production and release of cytolytic toxins is a critical aspect for the pathogenicity of many bacterial pathogens. In this study, we demonstrate a role for wall teichoic acids, molecules that are anchored to the peptidoglycan of the bacterial cell wall, in the release of toxins from *S. aureus* cells into the extracellular environment. Our findings suggest that this effect is mediated by a gradient of electrostatic charge which the presence of the negatively charged WTA molecules create across the cell envelope. This work brings an entirely new aspect to our understanding of the cytotoxicity of *S. aureus* and demonstrates a further means by which this major human pathogen can adapt its pathogenic capabilities.

**KEYWORDS** *Staphylococcus aureus*, cytolytic toxins, wall teichoic acids

Address correspondence to Ruth C. Massey, ruth.massey@bristol.ac.uk.

The authors declare a conflict of interest. M.J.A. and R.A. are shareholders of Integrated Biotherapeutics, Inc. but the company played no role in the direction of the research or interpretation of the results.

*S*taphylococcus aureus is a Gram-positive human pathogen that can cause a wide array of diseases, ranging from skin and soft tissue infections to bloodstream infections (1, 2). The ability to cause such a wide range of infections is determined by the

impressive array of virulence factors that *S. aureus* can produce, including adhesins, immune evasion factors, and cytolytic toxins (3–5). The cytolytic toxins produced by *S. aureus* can be classified as receptor-mediated toxins and non-receptor-mediated toxins (6). Alpha hemolysin (Hla), bicomponent toxins like leukocidins (e.g., LukSF, LukAB, and LukED), and gamma hemolysin (HlgAB and HlgCB) are receptor-dependent toxins and form stable multimeric pores upon interaction with a specific receptor on target cells (7, 8). Phenol-soluble modulins (PSMs) instead are small peptides that have detergent-like properties, with a broader array of cellular targets (9, 10).

*S. aureus* toxins generally carry out their function in the extracellular milieu, and hence, they need to cross the bacterial envelope to reach this location. As with most Gram-positive bacteria, the *S. aureus* cell envelope is composed of a cytoplasmic cell membrane and a peptidoglycan cell wall (11). Most proteins are translocated through the cytoplasmic membrane by the Sec system, although some proteins are translocated with specialized systems such as the SecA2/SecY2, Tat, or type VII secretion system (12, 13). A further system, named Pmt, is specifically used to translocate PSMs (14). In contrast to the cytoplasmic membrane, the cell wall is a porous polymer and secreted proteins are thought to passively diffuse through it (15–17). However, chemical and physical modifications can influence cell wall architecture and permeability, and the regulation of protein translocation through the cell wall is still an understudied process.

Teichoic acids are a major class of surface polymers found in Gram-positive bacteria that are named wall teichoic acids (WTA) if anchored to the peptidoglycan or lipoteichoic acids (LTA) if bound to the cytoplasmic membrane (18, 19). These phosphate-rich polymers create what has been described as a "continuum of negative charge" that extends from the bacterial membrane to the outermost layers of peptidoglycan (20). LTA and WTA share some functions and are involved in cell division, cell wall maintenance, and turnover (21–24). Despite their similarity, the biosynthesis of these polymers is performed by two distinct pathways. WTA biosynthesis is initiated by the TarO enzyme with the attachment of GlcNAc onto a lipid carrier (25, 26). WTA biosynthesis then continues in the inner leaflet of the cytoplasmic membrane with the contribution of several Tar enzymes, and the polymer is then flipped outward and attached to the peptidoglycan. In contrast, LTA polymerization happens on the outer leaflet of the cytoplasmatic membrane and involves the enzymes encoded by the *ltaA*, *ltaS*, and *ypfP* genes (27, 28). Recently, LTA has been associated with the secretion and sorting of the LukAB toxin (29). Unlike the other bicomponent toxins, LukAB is partially secreted and partially retained on the cell surface due to a sorting process that involves the cell membrane lipid lysyl-phosphatidylglycerol (LPG) and LTA. This demonstrates that toxin secretion can rely on complex mechanisms involving many diverse cell envelope macromolecules.

In previous work, a genome-wide association study (GWAS) identified an intergenic single nucleotide polymorphism (SNP) in a locus responsible for WTA synthesis as associated with changes in the cytolytic capacity of *S. aureus* (30). This SNP was located upstream of the *tarF* gene, which encodes a primase that attaches a single glycerol phosphate (GroP) onto the lipid-diphospho-GlcNAc-ManNAc-phosphoglycerol. This is subsequently used by TarL to add ribitol-5-phosphate repeats, which completes the polymerization (26, 31). Here, we explore the role of WTA in *S. aureus* toxicity. We found that a reduction in WTA production causes a reduction in the secretion of toxins and an increased abundance of toxins being retained within the cell. This mechanism did not involve a change in cell wall porosity, but instead, the toxins appear to be retained due to electrostatic interactions with membrane-bound molecules such as LTA. Our results suggest that toxin diffusion through the cell wall is a complex, stepwise mechanism that results from the interplay of a wide range of cell envelope components and includes the electrostatic gradient that spans from the membrane to the extracellular environment.

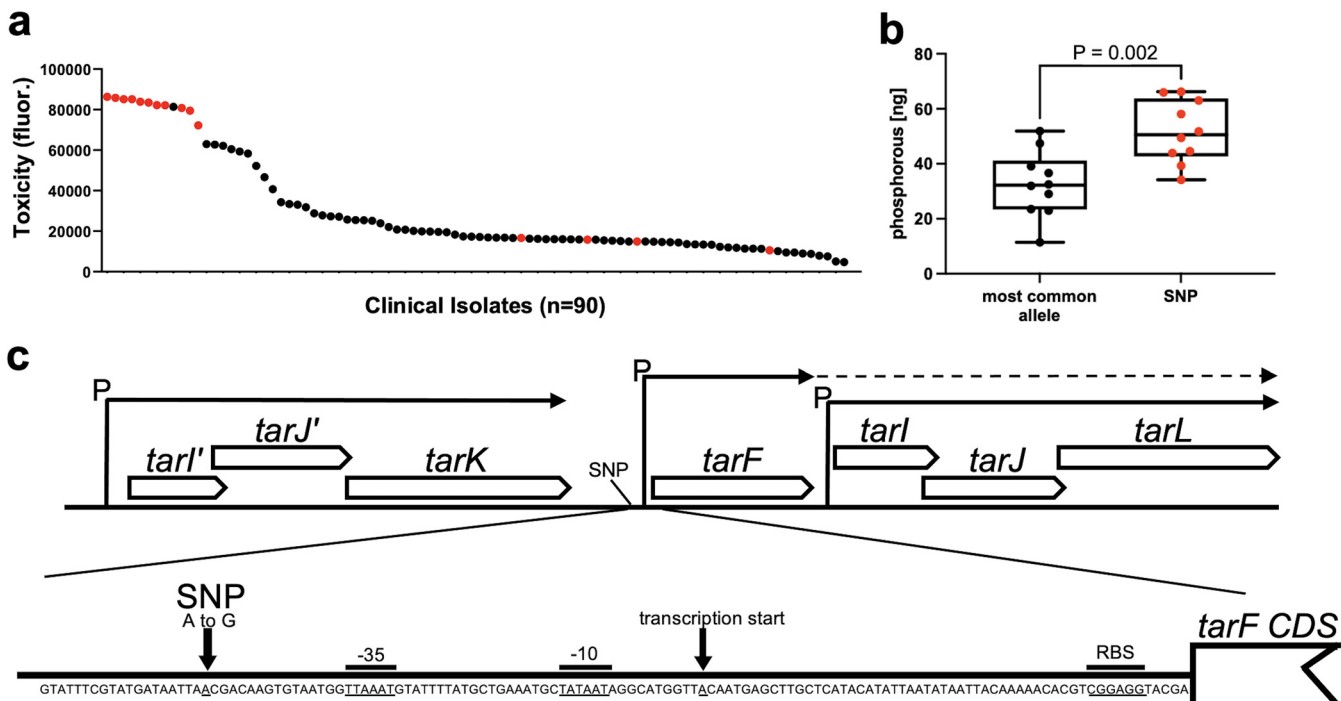

**FIG 1** A SNP in the *tar* locus of clinical *S. aureus* isolates is associated with higher levels of WTA and cytolytic activity. (a) The cytotoxicity of each clinical isolate is presented from the highest to the lowest, with the isolates containing the SNP in the *tar* locus indicated in red. (b) The SNP-containing clinical isolates produce significantly more WTA than an equivalent number of isolates that do not contain the SNP, as determined by the phosphorus content of the WTA extracts. The graph represents the amounts of phosphorus extracted from a 4-mL culture (at an $OD_{600}$ of 4). Each dot represents one isolate ($n = 10$ for each group), error bars represent the standard deviation, and statistical significance was determined by a *t* test. (c) Illustration of the position of the SNP within the *tar* locus; promoter features (P) were predicted using Softberry BPROM (http://www.softberry.com) (53). As indicated by the dashed line, there is evidence that the *tarF* promoter can read through and transcribe *tarIJL*, despite their having their own promoter (50).

## RESULTS

**Clinical isolates with a SNP within the *tarF* promoter region produce more WTA and more cytolytic toxins.** In previous work on a collection of 90 clinical methicillin-resistant *S. aureus* (MRSA) isolates, we identified an association between a single nucleotide polymorphism (SNP) in an intergenic region within the *S. aureus* WTA biosynthetic locus and cytolytic activity (30). To understand this association, we ranked the clinical isolates from most to least toxic and indicated which contained this SNP (Fig. 1a). Despite cytolytic activity being a polygenic trait, there is a clear association between the strains containing the SNP and high levels of cytolytic activity (of note is that of the four low-toxicity isolates that contain the *tarF* SNP, two also contain SNPs in the Agr locus, which explains their reduced toxicity, and the other two have SNPs in other putative toxicity-affecting loci that are currently undergoing analysis). To examine whether this intergenic *tar* SNP affected WTA production, we extracted WTA from 10 clinical isolates that contained the SNP and 10 isolates with the wild-type reference sequence (i.e., no SNP) and measured WTA abundance by quantifying the phosphorus levels of the WTA extracts, where the extracts from the isolates containing the SNPs had significantly higher levels of WTA than those with the reference *tarF* intergenic region (Fig. 1b). We did not observe any differences in the lengths of the WTA polymers from either set of isolates with or without the SNP (see Fig. S1 in the supplemental material). The SNP is located between the *tarK* and *tarF* genes and is just upstream of the *tarF* −35 site (Fig. 1c). In an attempt to determine the effect the SNP has on the transcription of the *tarF* gene, we performed quantitative reverse transcription PCR (qRT-PCR) on the clinical strains, with both 5 and 25 ng of cDNA, but we were unable to detect any *tarF* gene transcription. We also cloned this promoter region upstream of a green fluorescent protein (GFP) reporter system and detected no fluorescence with either the reference or SNP-containing promoter

region (Fig. S2). This suggests that *tarF* is transcribed by these clinical isolates at levels undetectable using these technologies.

**WTA production affects the cytolytic activity of *S. aureus*.** Although we were unable to verify a direct effect of the SNP, our analysis of clinical isolates suggests that mutations that affect WTA production also affect the cytolytic activity of *S. aureus*. To examine this further, we used a set of isogenic mutants of the MRSA strain LAC, which were selected because they allowed us to examine the effect of a range of WTA production levels (i.e., high [LAC], intermediate [LAC *tarKF*::Tn], and none [LAC Δ*tarO*]) on the cytolytic activity of the bacteria. It is worth noting that many of the *tar* genes, including *tarF*, cannot be inactivated due to the toxic effect a build-up of the partially formed components of WTA has on the bacteria (32, 33). To demonstrate the effect the mutations had on WTA production, we extracted WTA from the cells, visualized the WTA on a tricine acrylamide gel with alcian blue staining, and quantified the level of WTA production using ImageJ (Fig. 2a and b).

To verify the effect that various levels of WTA production had on cytolytic activity, we incubated bacterial supernatant with cultured cells from THP-1, an immortalized monocyte progenitor cell line that is susceptible to the majority of cytolytic toxins produced by *S. aureus* (34). Both mutants were significantly less cytolytic than the wild-type strain in a manner that was complemented by expressing the respective *tar* gene from the inducible plasmid pRMC2 (Fig. 2c and d). The effect on cytolytic activity was greater in the strain producing no WTA than in the strain producing an intermediate level, suggesting that there is a dose-dependent WTA effect on the cytolytic activity of *S. aureus*.

**WTA production affects toxin localization and secretion.** To determine whether toxins are differentially produced and/or secreted in the absence of WTA, we performed a series of bacterial supernatant and whole-cell lysate extractions and Western blot analyses to quantify the level of cytolytic toxins produced by the bacteria. There was a range of effects on the toxins detected: the AB leukotoxin (LukAB), gamma hemolysin (HlgA), and alpha toxin (Hla) were less abundant in the bacterial supernatants of the Δ*tarO* mutant than the wild type (Fig. 3a), which explains the reduced cytolytic activity of this mutant. However, in the whole-cell lysates of the Δ*tarO* mutant Hla, LukS and LukF were more abundant than in the wild-type strain, while LukAB was less abundant and HlgA was undetectable (Fig. 3b). This suggests that the release of some of these toxins (i.e., Hla, LukS, and LukF) from the bacterial cells is less efficient in the absence of WTA. Why the abundance of LukAB was lower in both the whole-cell lysate and supernatant of the Δ*tarO* mutant is as yet unclear and under investigation. We also examined the location of the phenol-soluble modulins (PSMs). These small surfactant-like toxins migrate ahead of the dye front as a single band of approximately 2 to 3 kDa on SDS-PAGE gels, which we have verified by mass spectroscopy (Table S1). A similar apparent trapping of the PSMs inside the cells was observed for both Δ*tarO* and *tarKF*::Tn strains (Fig. 3c and d).

To further examine where the toxins are being trapped within the bacterial cells, we separated the cell walls from the protoplasts with a focus on just Hla and PSM abundance, as these were the toxins with the clearest difference in their release from the bacterial cell between the wild type and WTA mutants, i.e., they were more abundant in the whole-cell extract of the mutants and less in the supernatant. In doing so, we found an increase in abundance of both Hla and the PSMs in both cellular compartments (Fig. 3e and f).

**Passive diffusion of toxin-sized molecules through the cell wall is not impaired in the absence of WTA.** In previous studies, the inactivation of TarO is reported to have a major effect on the architecture of the cell envelope (21, 35, 36). We confirmed this by transmission electron microscopy, where we also saw major morphological changes when no WTA was produced but also an increase in cell wall thickness when there was an intermediate level of WTA produced relative to the wild-type strain (Fig. 4a). We therefore hypothesized that passive diffusion of the cytolytic toxins through the cell wall may be inhibited by the different morphology of the cell wall of the WTA mutants. To examine whether WTA affect diffusion through the cell wall, we isolated murein sacculi

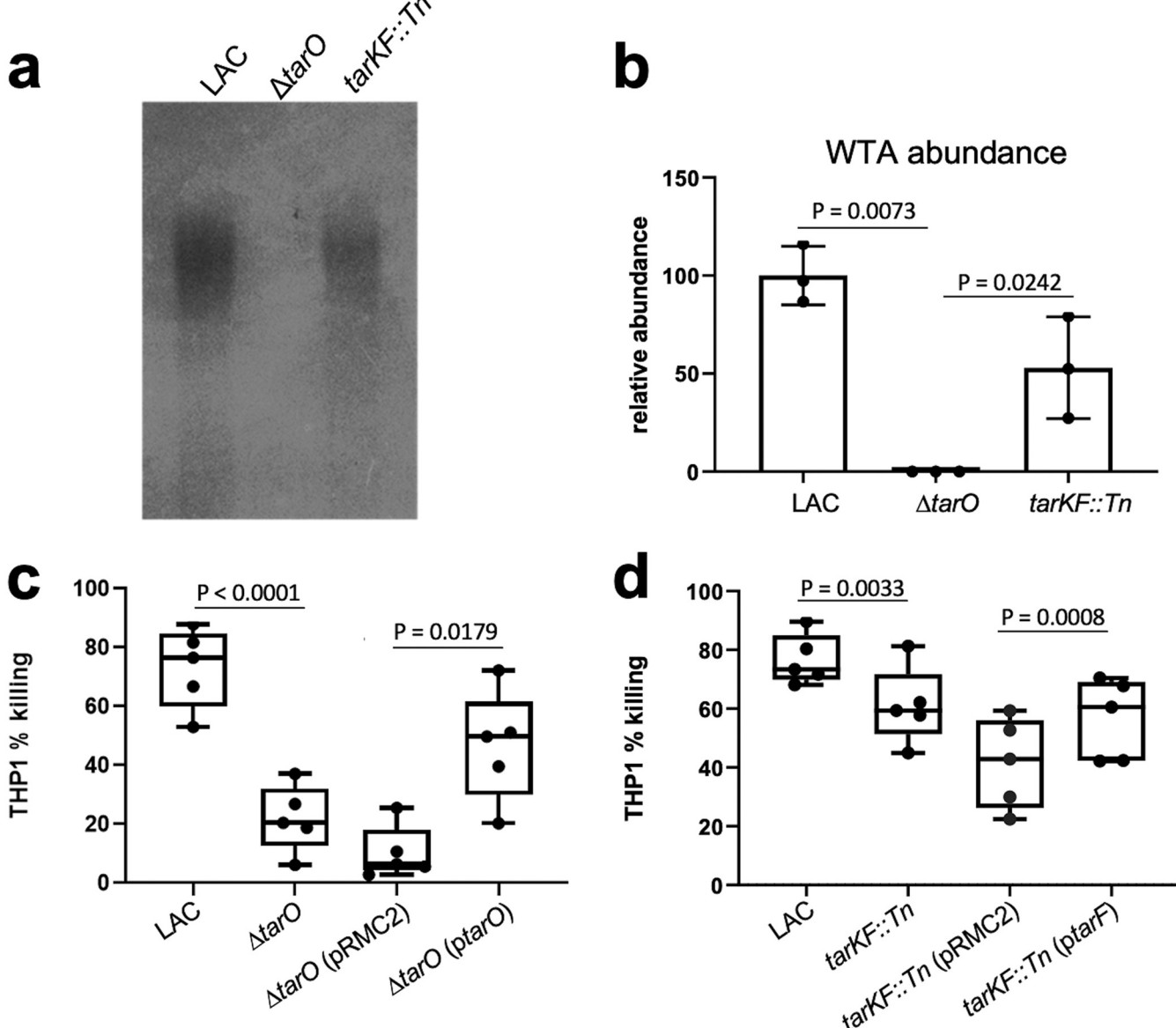

**FIG 2** WTA production affects the cytolytic activity of *S. aureus*. (a and b) WTA gels and ImageJ quantification, verifying the differing levels of WTA produced by the wild-type LAC strain, an isogenic mutant of LAC with no WTA (LAC Δ*tarO*), and a mutant of LAC with intermediate WTA (LAC *tarFK*::Tn). Each dot represents one biological replicate ($n = 3$), error bars represent the standard deviation, and statistical significance was determined by an unpaired *t* test. (c and d) THP-1 cell lysis upon incubation with supernatants harvested from bacterial overnight cultures. The toxicity of the isogenic mutant set demonstrates that lower levels of WTA production affect the cytolytic activity of *S. aureus*. This effect was complemented by expressing the respective genes from the pRMC2 plasmid. Each dot represents one biological replicate ($n = 5$), error bars represent the standard deviation, and statistical significance was determined by a paired *t* test.

of the wild-type and mutant strains and incubated these overnight with fluorescently labeled 40-kDa dextran, allowing them to reach an equilibrium. These dextran-filled sacculi were then diluted 1/50 in water to reduce the concentration of the extracellular dextran, and the rate of diffusion of the dextran from the sacculi was measured over a period of 30 min. The rate of diffusion of the dextran was equivalent across the strains, suggesting that the changes to the thickness of the cell wall in the absence of WTA is unlikely to be the explanation for the increase in the amount of toxins trapped within the cell wall of the WTA mutants (Fig. 4b).

**WTA-associated electrostatic charge across the cell wall affects the release of cytolytic toxins from the bacterial cell.** The cell envelope of *S. aureus* is predominantly negatively charged, due to the presence of negatively charged molecules such as the phospholipids in the membrane, membrane-bound lipoteichoic acids (LTA), and

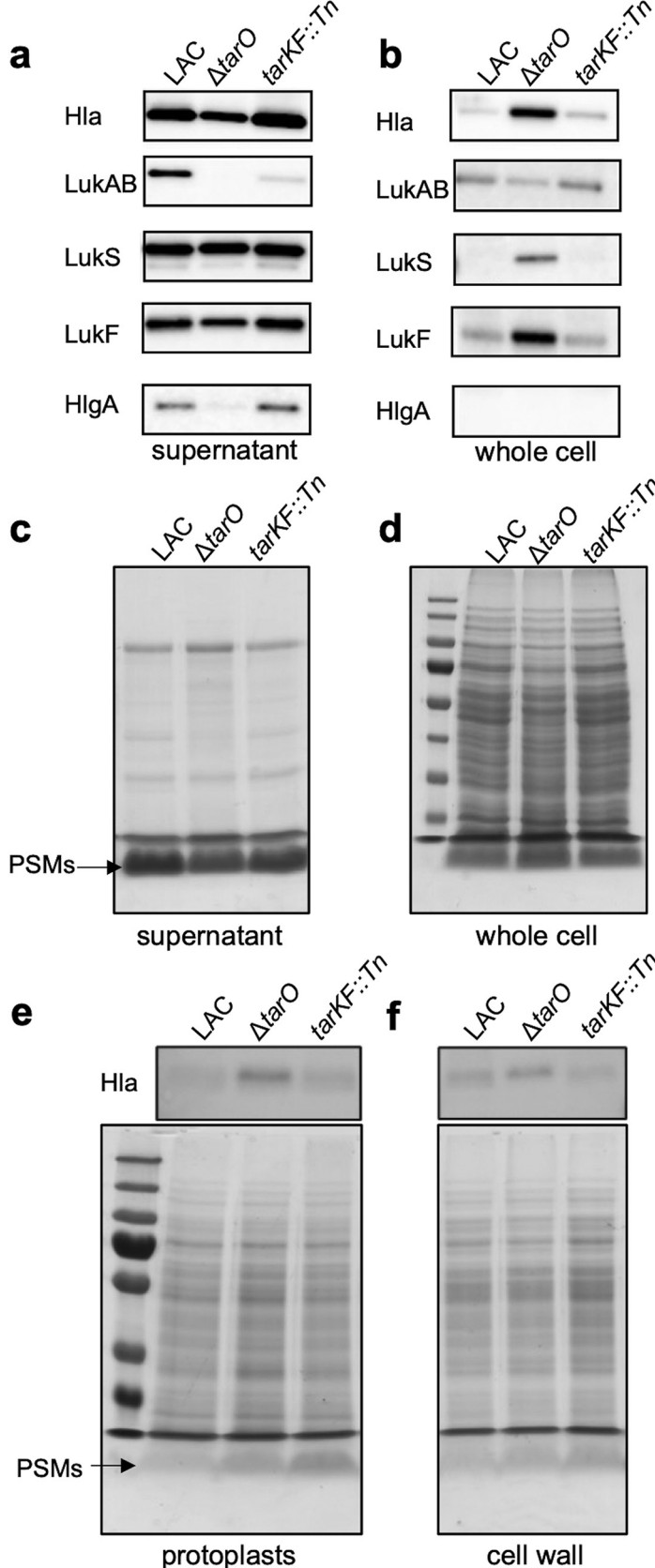

**FIG 3** WTA production affects the release of toxins from the *S. aureus* cell. (a and b) Western blotting detects toxin abundance in both intra- and extracellular extracts of the isogenic set of WTA-producing

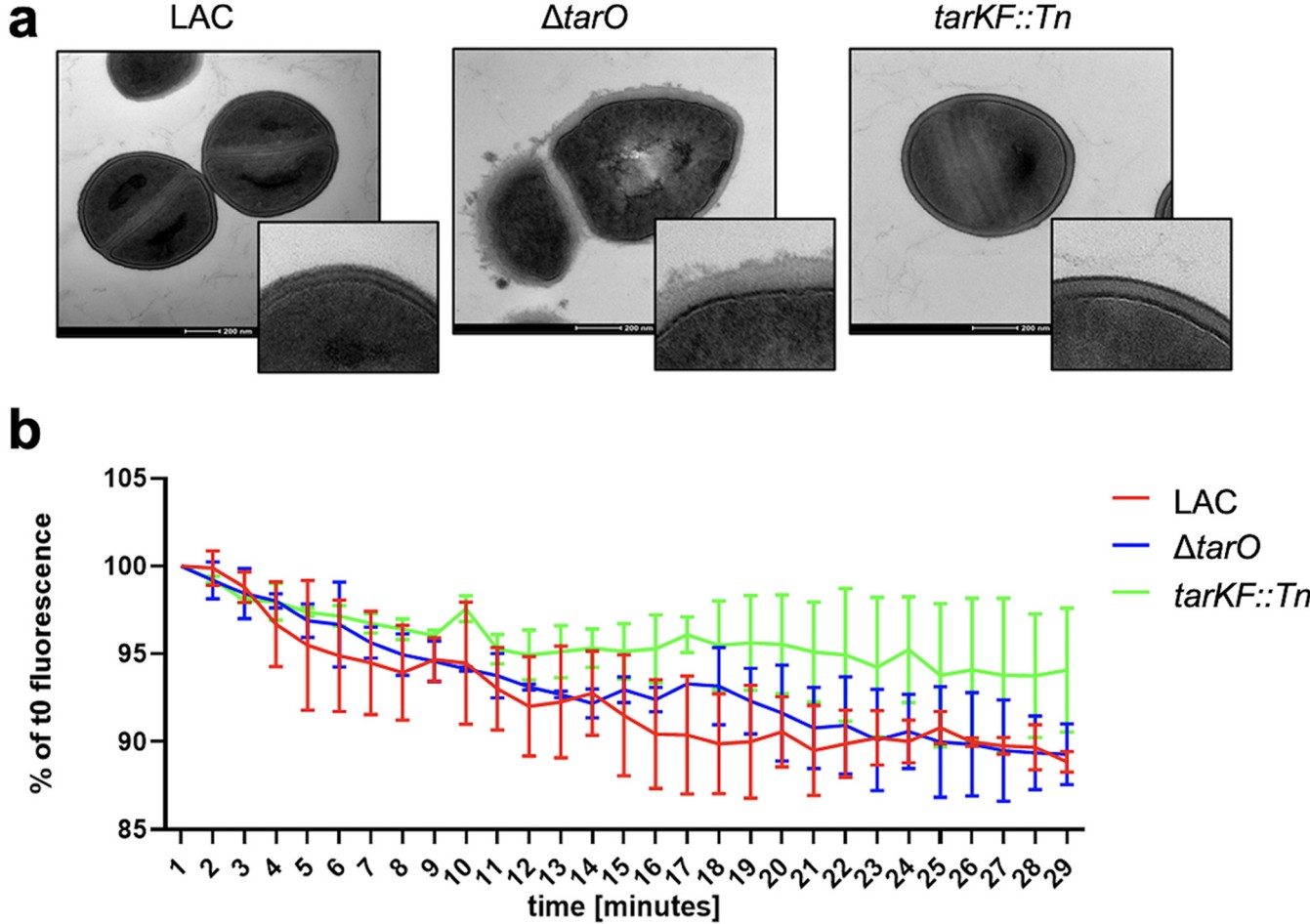

**FIG 4** Alterations in the architecture of the cell wall in the absence of WTA does not affect diffusion through it. (a) Transmission electron microcopy images demonstrating the changes that differing levels of WTA production make to the *S. aureus* cell wall. (b) WTA production levels do not affect dextran diffusion from the murein sacculi of the *S. aureus* strains producing various amounts of WTA. Murein sacculi were incubated overnight with labeled dextran, allowing their diffusion into the sacculi. The samples were then diluted in water, removing the dextran from the external environment of the sacculi, allowing the diffusion of the dextran back outside of the sacculi. The dextran diffusion was determined by measuring the fluorescence reduction of the sacculi over time. The graph represents the mean result of two independent biological repeats, and the bars represent the standard deviation.

cell wall-bound WTA. On the other hand, cytolytic toxins are predominantly positively charged (Table S2). This led us to hypothesize that these toxins may not be released from WTA mutant cells due to the changes in electrostatic forces caused by the lack of WTA, where the predominantly negatively charged WTA molecules may be involved in drawing the positively charged toxin away from the membrane and out toward the extracellular environment. To test this, we incubated TarO mutant cells in increasing concentrations of NaCl, where the increasing ionic strength should reduce any electrostatic interactions that may be occurring between the toxins and the negatively charged molecules within the membrane. This resulted in a decrease in the abundance of both Hla and the PSMs within the cells and a corresponding increase in the levels of these toxins in the supernatant, suggesting that in the absence of WTA, toxins are retained within the bacterial cells due to electrostatic interactions (Fig. 5a and b). We

**FIG 3** Legend (Continued)

strains, demonstrating that some toxins are not released from the bacterial cells. (c and d) PSM production of the isogenic set of WTA-producing strains, demonstrating that some of the PSMs are not released from the bacterial cells. (e and f) Western blotting and PSM gels detect the abundance of Hla and the PSMs, showing that they are trapped within both the cell wall and the protoplast of the *S. aureus* cells. Full-length gels and Western blots with replicates can be found in Fig. S3 in the supplemental material.

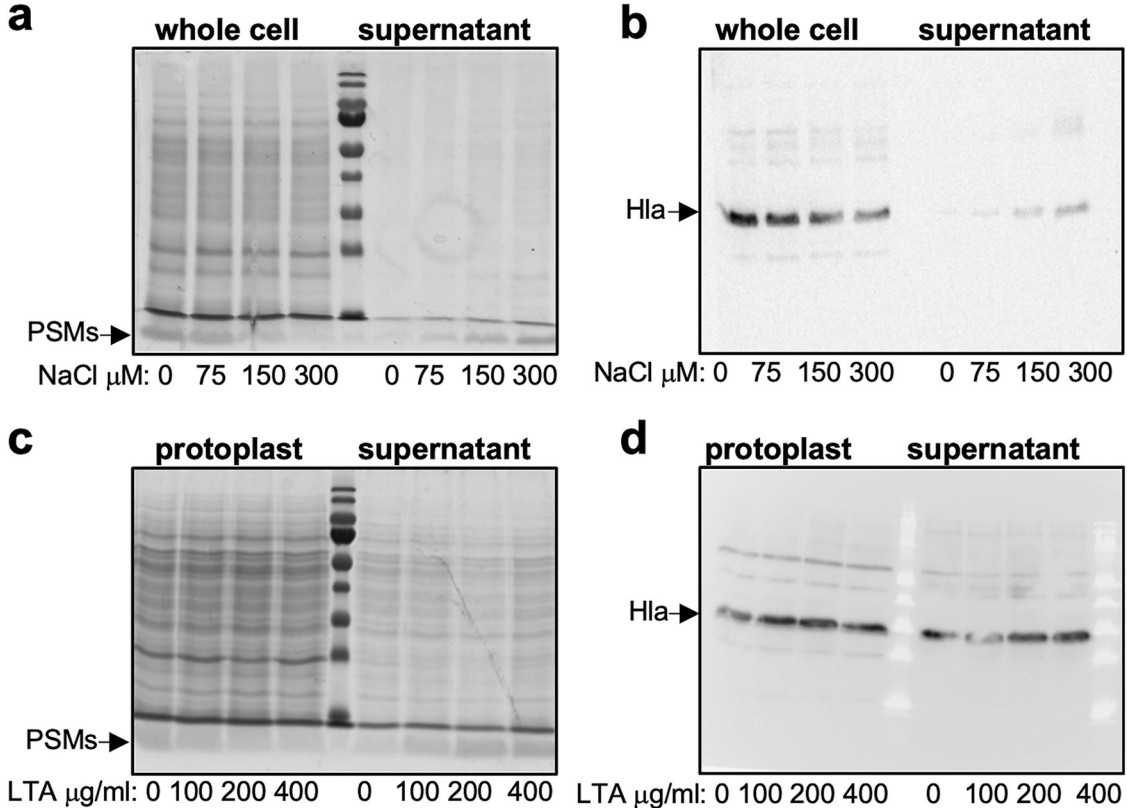

**FIG 5** In the absence of WTA, the PSMs are retained within the cells due to electrostatic interactions. (a and b) The PSMs and Hla of the TarO mutant moved from the cells into the supernatant when incubated in increasing concentrations of NaCl. (c) The PSMs of the TarO mutant moved from the protoplast into the supernatant when incubated with increased concentrations of soluble LTA. (d) Hla was rapidly released from the TarO protoplasts, and therefore any effect of Lta incubation on Hla release was not detectable.

then speculated that toxins might be held within the cell envelope by interacting with negatively charged LTA that sit in the bacterial membrane, so we tested whether purified soluble LTA could release either Hla or the PSMs from protoplasts. As with the increasing level of NaCl, with increasing concentrations of LTA, we observed a reduction in the abundance of the PSMs associated with the protoplast and an increase in the abundance of the PSMs released into the supernatant (Fig. 5c). However, Hla was immediately released from the protoplast in this assay such that the effect of the addition of LTA could not be determined (Fig. 5d). These findings further strengthen our hypothesis that electrostatic interactions between the positively charged toxins and negatively charged molecules in the cell membrane cause the retention of the toxins in the absence of WTA.

## DISCUSSION

The production and release of cytolytic toxins is a polygenic trait that begins with the sensing by the bacteria of the external environment for conditions in which the production of toxins will be most beneficial, through a complex regulatory cascade that leads to the transcription of the toxin genes (37–40). Once translated, the toxins are secreted through the cytoplasmic membrane, and up until now, their movement from the outer leaflet of the membrane was considered a passive diffusion process through the mesh-like structure of the peptidoglycan of the cell wall. However, in recent findings, a role for specific macromolecules in the cell envelope in the release of cytolytic toxins has been described. Leukocidin AB (LukAB) has been found to be retained as discrete foci in two distinct compartments of *S. aureus* cells: membrane-proximal and surface-exposed compartments, where LTA was found contribute to this

LukAB deposition and release (29). In addition to this, we here demonstrate that features such as electrostatic charge associated with key cell wall molecules such as WTA also play a critical role in the movement of toxins through the cell wall and their release into the extracellular environment. It is interesting to note that not all toxins were affected by the loss of WTA (Fig. 2a and b); why this is and how these other toxins traverse the cell wall are not yet understood, demonstrating how much we have yet to learn about this critical pathogenic trait.

A further level of complexity, which has not been addressed in this study, is represented by WTA modifications. The D-alanyl-lipoteichoic acid (DLT) pathway is responsible for incorporation of D-alanine into LTA and WTA, adding positive charges to these polymers and conferring zwitterionic features to teichoic acids. These modifications likely have an impact on the electrostatic interactions highlighted in this study, and it is tempting to speculate that they represent a further layer of the complexity by which *S. aureus* regulates and controls its toxin secretion.

Several studies have demonstrated how WTA abundance and WTA modifications mediate staphylococcal virulence (41, 42). In particular, increased WTA production has been shown to promote abscess formation, through an increased host immune response (41), while WTA glycosylation has been shown to have an impact on adherence to host epithelium and to govern nasal colonization (42). Despite this importance for survival *in vivo*, the bacteria can survive *in vitro* without WTA as long as a gene upstream of the WTA biosynthetic pathway, such as *tarO*, is inactivated (32, 33). Given this gene's importance, it is therefore interesting that we were unable to detect the transcription of the *tarF* gene using either a GFP fusion or qRT-PCR. It is possible that this is an *in vitro* effect, and if we had had access to *in vivo* samples, we may have been able to detect transcription and confirmed the effect that the SNP identified among our clinical isolates (Fig. 1c) has on the transcription of *tarF*. An alternative explanation relates to the negative effect the inactivation of other *tar* genes involved in the later stages of WTA biosynthesis has on the bacteria. Given the toxic effect on the bacteria that either these gene products and/or partially formed WTA subunits have when imbalanced, it is likely that their expression would be tightly controlled. It is therefore possible that the window of time in which the transcription of these genes is detectable is limited and was missed by us under these experimental conditions. It might also be the case that these genes were transcribed at a level too low to be detected by the techniques used here, but understanding whether any of these possibilities explains why we did not detect *tarF* transcripts requires an in-depth molecular investigation.

WTA is critical to the ability of *S. aureus* to colonize asymptomatically and to cause disease, and here, through the use of a functional genomic approach, we have described a further virulence mechanism regulated by WTA production and uncovered a hitherto unknown aspect of the cytotoxicity of a major human pathogen. Considering this alongside other recent findings, we propose that the movement of cytolytic toxins from the outer leaflet of the cytoplasmic membrane to the extracellular environment is a complex and largely opaque process involving the activity and physical presence of a number of cell wall-associated macromolecules. It is a research area that warrants further detailed investigation, given the potential that the blocking of such processes might bring to the development of novel therapeutic strategies.

## MATERIALS AND METHODS

**Bacterial strains and growth conditions.** The *S. aureus* laboratory strains used in this study are listed in Table 1. The plasmids used in this study are listed in Table 2. A list of the clinical strains, including information on whether they contained the *tar* intergenic SNP and their toxicity level as determined previously (30), is given in Table S3 in the supplemental material. Tryptic soy broth (TSB) and tryptic soy agar (TSA) were used for all *S. aureus* strain cultures. Strains containing transposon insertions were cultured on TSA plates with the addition of erythromycin (5 $\mu$g/mL). Strains containing pRMC2 plasmid were selected on chloramphenicol (10 $\mu$g/mL), while anhydrous tetracycline (100 ng/$\mu$L or 200 ng/$\mu$L) was added to the medium for the activation of the inducible promoter.

**Bacterial genetic manipulation.** The transposon insertion from the 95E07 strain (30) was transferred to LAC through phage transduction using phage Φ11, generating the strain LAC *tarKF*::Tn. Correct transposon insertion was verified through colony PCR using the tarKF_int_F and tarF_R primers. For

**TABLE 1** Strains used in this study

| Strain name | Description[a] | Reference |
|---|---|---|
| LAC | USA300; CA-MRSA, type IV SCC*mec* | 51 |
| LAC Δ*tarO* | *tarO* KO mutant in LAC | 52 |
| LAC Δ*tarO* (p*tarO*) | *tarO* KO mutant in LAC complemented with the *tarO* gene cloned into pRMC2 plasmid | This study |
| 95E07 | JE2 strain with a transposon insertion in the *tarK tarF* intergenic region, Tn::298351 | 30 |
| LAC *tarKF*::Tn | LAC strain with transposon insertion in the *tarK tarF* intergenic region, Tn::298351 (this is 42 nucleotides upstream of *tarF* ORF) | This study |
| LAC *tarKF*::Tn (p*tarF*) | LAC strain with transposon insertion in the *tarK tarF* intergenic region complemented with the *tarF* gene cloned into pRMC2 plasmid | This study |
| LAC *tarKF*::Tn (pRMC2) | LAC strain with transposon insertion in the *tarK tarF* intergenic region, transformed with empty pRMC2 vector | This study |

[a]CA-MRSA, community acquired MRSA; SCC*mec*, staphylococcal cassette chromosome *mec* element; KO, knockout; ORF, open reading frame.

complementation experiments, the *tarF* gene was amplified using JE2 genomic DNA and KAPA HiFi polymerase (Roche) with the primers tarF_F and tarF_R, while the *tarO* gene was amplified using the primers tarO_F and tarO_R. tarF_F and tarO_F primers contained the KpnI restriction site at the 5′ end, while tarF_R and tarO_R included a SacI restriction site. The PCR products were cloned into the pRMC2 plasmid, using KpnI and SacI restriction enzymes and T4 ligase (NEB) to make the p*tarO* and p*tarF* plasmids. These were first transformed via electroporation into the RN4220 strain and eventually transformed into the LAC mutants to complement LAC Δ*tarO* and LAC *tarKF*::Tn mutations.

**WTA extraction and analysis.** WTA extraction was performed as described previously (43). Briefly, the bacteria were grown overnight, and any difference in growth was quantified in a spectrophotometer at an optical density at 600 nm (OD$_{600}$) and adjusted using fresh broth. A 4-mL volume of the cultures normalized to an OD$_{600}$ of 4 was centrifuged and washed with buffer 1 (50 mM MES [morpholineethanesulfonic acid], pH 6.5). The samples were boiled for 1 h in SDS containing buffer 2 (4% [wt/vol] SDS, 50 mM MES, pH 6.5) to remove all LTA and contaminating lipids. Samples were then washed sequentially in buffer 1, buffer 2, buffer 3 (2% NaCl, 50 mM MES, pH 6.5), and buffer 1 again to remove all lipids and residual SDS. Proteins were removed by digestion with proteinase K for 4 h at 50°C in digestion buffer (20 mM Tris-HCl, pH 8.0, 0.5% [wt/vol] SDS). Samples were washed in buffer 3 and then three times in Milli-Q H$_2$O. Samples were then incubated for 16 h in 0.1 M NaOH to release the WTA from the peptidoglycan. The samples were centrifuged, and the supernatants were neutralized with 1 M Tris-HCl, pH 7.8. The WTA extraction samples were stored at −20°C before proceeding with the following analyses. Phosphorus amounts in WTA extraction samples were measured using the colorimetric ammonium molybdate and ascorbic acid method as previously described (44). A standard curve was produced with phosphorus standards, and the total amounts of phosphorus extracted from the bacterial cultures were calculated. WTA acrylamide gels were run as previously described (43) in Tris-Tricine buffer (1 M Tris, 1 M Tricine, pH 8.2). The gels were separated at 4°C and 40 mA with constant stirring until the dye front reached the bottom. The gels were washed three times in Milli-Q H$_2$O and then stained overnight with alcian blue solution (1 mg/mL alcian blue, 3% acetic acid). Gels were then destained in Milli-Q H$_2$O until the WTA became visible.

**RNA extraction and qRT-PCR.** *S. aureus* clinical isolates were inoculated at an OD of 0.05 in a flask, and the cultures were incubated at 37°C with shaking at 180 rpm. Samples for RNA extraction were collected from 1 mL of *S. aureus* isolates grown to exponential phase (OD$_{600}$ of 2) and stabilized using RNAprotect bacterial reagent (Qiagen). RNA extraction was performed using a Quick-RNA fungal/bacterial miniprep kit, in accordance with the manufacturer's instructions. The extracted RNA was treated with a TURBO DNA-free kit (Thermo Fisher), and reverse transcription was performed using the qScript cDNA synthesis kit (Quantabio). Real-time PCR was performed using a KAPA SYBR fast qPCR kit (Kapa Biosystems), using the primers tarF_RT_F (CTTTCATGGTAAACAATACAGCG) and tarF_RT_R (TGTTGGAATATGTGTTCATATCC). *gyrB* was used as the housekeeping gene, using the primers gyrB_RT_F (GGTGACTGCATTGTCAGATGTAAAC) and gyrB_RT_R (CTGCTTCTAAACCTTCTAATACTTGTATTTG). Both assays were validated using JE2 genomic DNA, showing reaction efficiencies of 98.98% for *tarF* primers and 98.48% for *gyrB* primers. A first reaction was performed using 5 ng of cDNA, which gave threshold cycle ($C_T$) values higher than 30 and close to the negative controls for *tarF*, while the $C_T$ values for *gyrB* were approximately 22. The real-time PCR was then repeated with 25 ng of cDNA, resulting again in $C_T$ values higher than 30 and close to the negative control for the *tarF* transcript.

**THP-1 toxicity assay.** The monocytic THP-1 cell line (ATCC TIB-202) was used as previously described (34, 45). Cells were cultivated in RPMI 1640 supplemented with heat-inactivated fetal bovine serum (10%), ʟ-glutamine (1 M), penicillin (200 U/mL), and streptomycin (0.1 mg/mL) in a humidified incubator

**TABLE 2** Primers used in this study

| Primer name | Sequence |
|---|---|
| tarKF_int_F | TGTTGGCATATATTACTTGCCATG |
| tarF_F | TATA<u>GGTACC</u>TTACAAAAACACGTCGGAGG |
| tarF_R | ATAT<u>GAGCTC</u>ATTTAAGTTATCACTTAAAAATCGTTTGG |
| tarO_F | TATA<u>GGTACC</u>TTAATATCGATGAAGGTGAATAAATG |
| tarO_R | ATAT<u>GAGCTC</u>AGCTATGCTTTCATTCCCTATTC |

at 37°C with 5% $CO_2$. For toxicity assays, cells were harvested by centrifugation and resuspended to a final density of $1 \times 10^6$ to $1.5 \times 10^6$ cells/mL in tissue-grade Hanks' balanced salt solution (HBSS), typically yielding 95% viability. Bacterial supernatants harvested from *S. aureus* overnight cultures (18 h) were diluted to 10 to 30% in TSB and mixed 1:1 with THP-1 cells incubated for 12 min at 37°C. THP-1 cell death was measured by a trypan blue exclusion assay. Three technical replicates were performed for each biological replicate, and independent biological replicates were performed on different days.

**Cell fractionation and protein analysis.** Whole-cell lysates were prepared from overnight bacterial cultures, normalized to the same OD. Samples were centrifuged, and the supernatants were stored at −20°C. Cell pellets were washed once in phosphate-buffered saline (PBS) and then treated with lysostaphin to digest the cell wall. SDS was added to the suspensions at a final concentration of 2%, and the samples were boiled for 20 min. The samples were stored at −20°C. Cell wall and protoplast fractions were obtained from overnight cultures. Cell pellets were washed in TSM buffer (50 mM Tris-HCl, pH 7.5, 0.5 M sucrose, 10 mM $MgCl_2$) and treated with lysostaphin in TSM buffer for 15 min at 37°C. Samples were centrifuged at 8,000 rpm, and supernatants containing the cell wall fraction were stored at −20°C. The pellets, containing the protoplasts, were resuspended in PBS–2% SDS and boiled for 20 min, and then samples were stored at −20°C. For protein gels, samples were mixed with 2× SDS loading dye, boiled for 10 min, and loaded onto 12% acrylamide gels. Either the gels were stained with Coomassie blue for PSMs and total protein was visualized or the gels were transferred to a nitrocellulose or polyvinylidene difluoride (PVDF) membrane for Western blotting. Transfer was performed using either a Trans-Blot Turbo transfer system (Bio-Rad) or an iBlot2 transfer device (Life Technologies). The membranes were blocked either overnight in PBS-Tween-milk (10%) or for 10 min in StartingBlock T20 (TBS). The membranes were incubated with toxin-specific antibodies (1:2,000, 1 $\mu$g/mL) in either PBS-Tween-milk (3%) for 1 h or in StartingBlock T20 (TBS) overnight. Membranes were washed and incubated with horseradish peroxidase-conjugated secondary antibodies for 1 h. Proteins were detected by using either an Opti-4CN detection kit (Bio-Rad) or an ECL detection kit (Cytiva). The LukAB dimer antibodies were generated from purified dimers as described previously (46). The Hla antibodies are monoclonal (IBT Bioservices; catalog no. 0210-005). Unique peptide-specific rabbit polyclonal antibodies were generated targeting unique sequences of HlgA and LukF and LukS (from GenScript). Detailed Western blot protocols are given in our previous publications (47, 48).

**PSM band identification.** Bacteria (JE2) were cultured for 18 h in 5 mL TSB and centrifuged to separate the cell pellet from the supernatant. Proteins in the supernatant fraction were concentrated via trichloroacetic acid (TCA) precipitation. Briefly, 1 volume of cold (4°C) TCA (VWR) was mixed with 4 volumes of supernatant and incubated on ice for 60 min. Precipitated proteins were pelleted by centrifugation and washed in ice-cold acetone three times. Protein pellets were resuspended in 50 $\mu$L 8 M urea. Protein samples were then mixed with blue protein loading dye (NEB) boiled for 10 min and separated on a 10% SDS-PAGE gel for 50 min at 140 V. The gel was stained with Quick Coomassie (Generon) for 2 h and destained overnight in deionized water. The band below the dye front, corresponding to less than 5 kDa, was excised, stored in deionized water, and delivered to the Bristol Proteomics Facility, where the band was subjected to in-gel proteolytic digestion using an automated DigestPro. The resulting peptides were analyzed by matrix-assisted laser desorption ionization–time of flight tandem mass spectrometry (MALDI-TOF MS/MS) using a Bruker Daltonics ultrafleXtreme 2 mass spectrometer.

**Dextran diffusion assay.** The dextran diffusion assay was performed as previously described, with few adaptations of the protocol (49).

Briefly, murein sacculi were extracted by following the same protocol of WTA extraction, with exclusion of the final NaOH treatment. Sacculi were incubated overnight in a solution of 5 $\mu$M 40-kDa dextran labeled with Texas Red. This allows the dextran molecules to diffuse into the sacculi, reaching a homogeneous concentration with the solution. Samples were then diluted 1:50 in water, thus decreasing the dextran outside of the sacculi. At this point, the dextran within the sacculi tends to diffuse out to reach equilibrium. The dextran diffusion through the cell wall toward the external solution was followed, measuring the fluorescence of the sacculi by fluorescence-activated cell sorting (FACS) for 30 min. Mean fluorescence at 1-min intervals was used to compare different samples.

**Release of toxins by NaCl and LTA.** Cells from overnight cultures were pelleted by centrifugation and subsequently washed twice in PBS. The samples were resuspended in PBS with increasing concentrations of NaCl and incubated at 37°C and 180 rpm for 1 h. The cell suspensions were then centrifuged, and the supernatants were stored at −20°C. The pellets, containing the cells, were treated as described previously for whole-cell lysate preparation. Protoplasts were isolated as described previously for cell fractionation. The protoplasts were resuspended in TSM buffer with increasing concentrations of LTA and incubated at 37°C with shaking at 180 rpm for 10 min. The samples were then centrifuged for 5 min at 5,000 rpm, supernatants were stored at −20°C, and pellets were resuspended in TSM buffer–2% SDS.

**Electron microscopy.** Cells from overnight cultures were pelleted by centrifugation, washed once in PBS, and resuspended in 2.5% glutaraldehyde in 0.1 M sodium cacodylate buffer, pH 7.2. Fixed bacteria were delivered to the Wolfson Bioimaging Facility, University of Bristol, where they were dehydrated, embedded in resin, stained, and sectioned. Images were acquired using a FEI Tecnai 12 electron microscope.

**Statistical analysis.** Statistical analyses were performed using GraphPad Prism 8.4.

## SUPPLEMENTAL MATERIAL

Supplemental material is available online only.

**SUPPLEMENTAL FILE 1**, PDF file, 1.5 MB.

## ACKNOWLEDGMENTS

We thank Simon Foster, Laia Pasquina Lemonche, and Richard Daniel for helpful discussions of the data. We also acknowledge the assistance of Andrew Herman and Helen Rice for cell sorting, the University of Bristol Faculty of Life Sciences Flow Cytometry Facility, and Chris Neal, Mark Jepson, Judith Mantel, and Lorna Hodgson from the Wolfson Bioimaging Facility for their electron microscopy services.

M.J.A. and R.A. are shareholders of Integrated Biotherapeutics, Inc., but the company played no role in the direction of the research or interpretation of the results.

This work was funded by a BBSRC grant, a Wellcome Trust-funded Investigator award to R.C.M. (grant no. 212258/Z/18/Z), and a grant from National Institute of Infectious Diseases (NIAID) (R43AI136143) to R.A.

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
