## [Reviewer comments · Microbiology Spectrum]

Microbiology Spectrum

Wall teichoic acids facilitate the release of toxins from the surface of *Staphylococcus aureus*.

Tarcisio Brignoli, Edward Douglas, Seána Duggan, Olayemi Fagunloye, Rajan Adhikari, Javad Aman, and Ruth Massey

Corresponding Author(s): Ruth Massey, University of Bristol

Review Timeline:

Submission Date:

May 29, 2022

Accepted:

June 6, 2022

Editor: Christopher LaRock

Reviewer(s): The reviewers have opted to remain anonymous.

Transaction Report:

DOI: <https://doi.org/10.1128/spectrum.01011-22>

June 6, 2022

Prof. Ruth Catherine Massey
University of Bristol
School of Cellular and Molecular Medicine
Biomedical Sciences Building
University Walk
Bristol BS8 1TD
United Kingdom

Re: Spectrum01011-22 (Wall teichoic acids facilitate the release of toxins from the surface of *Staphylococcus aureus*.)

Dear Prof. Ruth Catherine Massey:

Your manuscript was transferred with recommendations from expert reviewers in the field. Your comprehensive responses and edits satisfy each of their comments and no additional review will be needed. Your manuscript has been accepted, and I am forwarding it to the ASM Journals Department for publication. You will be notified when your proofs are ready to be viewed.

Sincerely,

Christopher LaRock
Editor, Microbiology Spectrum
